# COIN: Counterfactual Image Generation for Visual Question Answering Interpretation

**DOI:** 10.3390/s22062245

**Published:** 2022-03-14

**Authors:** Zeyd Boukhers, Timo Hartmann, Jan Jürjens

**Affiliations:** 1Faculty of Computer Science, University of Koblenz-Landau, 56070 Koblenz, Germany; tihartmann@uni-koblenz.de (T.H.); juerjens@uni-koblenz.de (J.J.); 2Fraunhofer Institute for Software and Systems Engineering ISST, 44227 Dortmund, Germany

**Keywords:** ML interpretability, VQA, GAN, UXE

## Abstract

Due to the significant advancement of Natural Language Processing and Computer Vision-based models, Visual Question Answering (VQA) systems are becoming more intelligent and advanced. However, they are still error-prone when dealing with relatively complex questions. Therefore, it is important to understand the behaviour of the VQA models before adopting their results. In this paper, we introduce an interpretability approach for VQA models by generating counterfactual images. Specifically, the generated image is supposed to have the minimal possible change to the original image and leads the VQA model to give a different answer. In addition, our approach ensures that the generated image is realistic. Since quantitative metrics cannot be employed to evaluate the interpretability of the model, we carried out a user study to assess different aspects of our approach. In addition to interpreting the result of VQA models on single images, the obtained results and the discussion provides an extensive explanation of VQA models’ behaviour.

## 1. Introduction

Over the past years, the task of Visual Question Answering (VQA) has been widely investigated taking advantage of the development strides of Natural Language Processing (NLP) and Computer Vision (CV). A VQA model aims to answer a natural language question about the content of an image or one of the appearing objects. Due to the complexity of the task, VQA systems are still in the early stage of research and up to our knowledge, they are not integrated into any running system. One of the inherent problems of VQA systems is the reliance on the correlation between the question and answer more than the content of the image [1,2]. Furthermore, the available datasets are usually unbalanced w.r.t certain types of questions. In the VQAv1 dataset [3], for instance, simply answering “*tennis*” to any sports-related question without considering the image yields an accuracy of approximately 40%. This is because the dataset creators tend to generate questions about objects detectable in the image which make the dataset suffer from the so-called “*visual priming bias*”. For example, blindly answering “*yes*” to all questions starting with “*Do you see a...?*” without considering anything else yields approximately 90% accuracy in the VQAv1 dataset [2,4]. In practice, this bias is not distinctly perceivable because the users tend to ask similar questions related to the image or the appearing objects and they most likely know the correct answer. For more complex questions or when the user lacks the knowledge expertise of the questions or the image content (e.g., medical domain), it won’t be possible to capture the behaviour of the VQA system and whether it is biased. Therefore, it is important to interpret the result of these systems and find what caused the model to output an answer based on the image-question pair.

Although there is no uniform definition of “*Interpretability*”, researchers agree that the interpretability of ML models increases the users’ trust in ML systems. For VQA models, there exist a limited number of papers that investigate the interpretation of their models. Existing attempts include (i) the identification of visual attributes that are relevant to the question [5,6] and (ii) the generation of counterfactuals [2,7,8].

In the direction of (i), the method proposed by Zhang et al. [5] generates a heatmap over the input image to highlight the image regions that are relevant to the answer of the VQA system. However, as pointed out by Fernández-Loría et al. [9], this approach explains the system’s prediction but does not provide a sufficient explanation of its decision. They suggest instead that counterfactual explanations offer a more sophisticated way to increase the interpretability of an ML model because they reveal the causal relationship between features in the input and the model’s decision. For example, considering an ML model that classifies MRI images into *Malignant* or *Benign*, the generated heatmap can highlight the key Region Of Interest (ROI) of the model’s prediction. Although this heatmap answers the question *What did lead to this decision?*, it does not answer another important question: *Why did it lead to this decision?* Consequently, this approach does not provide insight into how the model would behave under alternative conditions. To provide more in-depth interpretability, generating a counterfactual image that is minimally different from the original one but leads to a different model’s output would indirectly answer the question *Why did the model take such a decision?* To the best of our knowledge, only three existing methods [1,7,8] aim at making VQA models interpretable by providing counterfactual images.

Chen et al. [1] introduce a method that generates counterfactual samples by applying masks to critical objects in the images or questions’ words. Similarly, Teney et al. [8] present a method that masks features in the images whose bounding boxes overlap with human-annotated attention maps. Finally, in their ongoing research, Pan et al. [7] propose a framework to generate counterfactual images by editing the original image such that the VQA system returns an alternative answer for a given question. Due to the complexity of the problem, the approach is restricted to colour-based questions. Given a tuple (Image, Question and the VQA’s answer), the approach first finds the question-critical object and then changes its colour so that the VQA system gives a different answer. However, this change is not limited to the question-critical object but all regions with similar colours are changed. Given that the main goal of interpreting VQA systems is to help the user understand the behaviour of the VQA model, the approach presented in [7] requires that the user understands the relationship between the image, the question and the answer.

As the user needs interpretation mainly when he lacks necessary knowledge to understand the relationship between the input and output, this paper aims to best interpret the output of VQA systems by generating counterfactual images that lead the VQA model to either (1) output a different answer or (2) deviate its focus on another region. Specifically, this paper aims to answer the following research questions:**RQ1:** How to change the answer of a VQA model with the minimum possible edit on the input image?**RQ2:** How to alter exclusively the region in the image on which the VQA model focuses to derive an answer to a certain question?**RQ3:** How to generate realistic counterfactual images?

To this end, we propose to extend the work proposed Pan et al. [7] by restricting the changes to the question-critical region. Specifically, this paper introduces an attention mechanism that identifies question-critical ROI in the image and guides the counterfactual generator to apply the changes on those regions. Moreover, a weighted reconstruction loss is introduced in order to allow the counterfactual generator to make more significant changes to the question-critical ROI than the rest of the image. For further improvement and future work, we made the entire implementation of the guided generator publicly available.

Following this section, Section 2 discusses the related works. Section 3 presents the proposed approach and Section 4 presents the conducted experiments and the obtained results that validate the effectiveness of the proposed approach. Finally, Section 5 concludes this paper and gives insight into future directions.

## 2. Related Work

This paper addresses the problem of interpreting the outcome of VQA systems. Therefore, we will review in this section the related works divided into three categories: (1) Interpretable Machine Learning, (2) Visual Question Answering (VQA) and (3) Interpretable VQA.

### 2.1. Interpretable Machine Learning

Throughout the past decades, the notion of interpretability increasingly gained attention by the Machine Learning (ML) community [10,11,12,13,14,15,16,17]. According to Kim et al. [13], interpretability is particularly important for systems whose decisions have a significant impact such as in healthcare, criminal justice and finance. Interpretability serves several purposes, including protecting certain groups from being discriminated against, understanding the effect of parameter and input variation on the model’s robustness and increasing the user’s trust in automated intelligent systems [11]. Therefore, a model is considered interpretable if it allows a human to consistently and correctly classify its outputs [13] and understand the reason behind the model’s output [10]. ML models such as decision trees are inherently interpretable, as they provide explanations during training or while the output is generated. However, most of the sophisticated ML models used nowadays such as Deep Neural Networks are not interpretable by nature and require post-hoc explanations [18].

One way to create such explanations is to use global surrogate models, where the aim is to approximate the prediction function *f* of a black-box and complex model (e.g., neural networks) to the best possible with a prediction function *g* such that *g* is the prediction function of an inherently interpretable model (e.g., decision trees or linear regression) [18,19]. Another way is to use local surrogate models, which individually explain the predictions of a trained ML model to have an overview of its behaviour [19]. Ribeiro et al. [20] propose a Local Interpretable Model-agnostic Explanations (LIME) which approximates the output of black-box models by examining how variations in the training data affect its predictions. Particularly, LIME permutes a trained black-box model’s training samples to generate a new dataset. Based on the black-box model’s predictions on the permuted dataset, LIME trains an interpretable model, which is weighted by the proximity of the sampled instances to the instance of interest.

Feature visualization is another direction to increase the interpretability of black-box models. The goal is to visualize the features that maximize the activation of a NN’s unit [19]. This direction takes advantage of the structure of NNs, where the relevance is backpropagated from the output layer to the input layer [16]. Mainly, most of the approaches under this direction are dedicated to image classification tasks by providing a saliency map that highlights the pixels relevant to the model’s output [18,21,22]. A common interpretability direction suggests generating example-based explanations for complex data distributions. The aim is to find prototypes from the training dataset that summarize the prediction of the model [18]. Although this approach can satisfy the user need for interpretation in simple tasks, it is not practical for most of the real-world data which are heavily complex and seldom contain representative prototypes [13]. Therefore, Kim et al. [13] propose to identify some criticism samples that deliver insights about those prototypes which is not covered by the model.

The problem of post-hoc interpretability methods is their incapability to answer how the model would behave under alternative conditions (e.g., different training data). Therefore, causal interpretability approaches aim at finding why did the model make a certain decision instead of another one or what would be the output of the model for a slightly different input [18]. Here, the goal is to extract causal relationships from the data by analysing whether changing one variable cause an effect in another one [23]. One way to achieve this is by finding a counterfactual input that affects the model’s prediction [24,25,26,27,28,29]. For instance, Goyal et al. [26] propose a method that identifies how a given image “*I*” could change so that the image classifier outputs a different class by replacing the key discriminative regions in “*I*” with pixels from an identified “distractor” image “$I_{0}$” that has a different class label. Pearl [30] suggests that generating counterfactuals allows for the highest degree of interpretability among all methods to explain black-box models.

### 2.2. Visual Question Answering (VQA)

Taking advantage of the remarkable advancement of Computer Vision, Natural Language Processing and Deep Neural Networks, several research works have addressed the task of VQA in the past years [1,3,4,31,32,33,34,35,36]. According to Antol et al. [3], VQA methods aim at answering natural language questions about an input image. This combination of image and textual data makes VQA a challenging multi-modal task that involves understanding both the question and the image [32]. The answer’s format can be of several types: a word, a phrase, a binary answer, a multiple-choice answer, or a “fill in the blank” answer [32,36].

In contrast to earlier contributions, recent VQA approaches aim at generating answers to free-form open-ended questions [37]. Agrawal et al. [3], for example, propose a system that classifies an answer to a given question about an image by combining a Convolutional Neural Networks (CNN)-based architecture to extract features from the image and Long Short-Term Memory (LSTM)-based architecture to process the question. This model, referred to as *Vanilla VQA*, can be considered as a benchmark for DL-based VQA methods [32]. Yang et al. [38] introduce *Stacked Attention Networks (SANs)* that uses CNNs and LSTMs to compute an images’ regions related to the answer based on the semantic representation of a natural language question. Similarly, Anderson et al. [34] propose to narrow down the features in images by using top-down signals based on a natural language question to determine what to look for. These signals are combined with bottom-up signals stemming from a purely visual feed-forward attention mechanism.

Despite the continuous advancements in VQA, several papers suggest that VQA systems tend to suffer from the language prior problem, where they tend to achieve good superficial performances but do not truly understand the visual context [1,4,33,39]. Specifically, Goyal et al. [4] found that in the VQAv1 dataset [3] blindly answering “*yes*” to any question starting with “*Do you see a...?*” without taking into account the rest of the question or the image yields an accuracy of 87%. To overcome this, they proposed a balanced dataset to counter language biases, such that for a given triplet (image *I*; question *Q*; answer *A*) from the VQAv1 dataset [3], humans were asked to identify a similar image $I^{\prime }$ for which the answer to question *Q* is different from *A*. Similarly, Zhang et al. [33] propose a balanced VQA dataset for binary questions, where for each question, pairs of images showing abstract scenes were collected so that the answer to the question is “*yes*” for one image and “*no*” for the other.

Moreover, Chen et al. [1] assume that existing VQA systems capture superficial linguistic correlations between questions and answers in the training set and, hence, yields low generalizability. Therefore, they propose a model-agnostic Counterfactual Samples Synthesizing (CSS) training scheme that aims at improving VQA systems’ visual-explainable and question-sensitive abilities. The CSS algorithm masks (i) objects relevant to answering a question in the original image to generate a counterfactual image and (ii) critical words to synthesize a counterfactual question. In the same context, Zhu et al. [39] propose a self-supervised learning framework that balances the training data but first, identifies whether a given question-image pair is relevant (i.e., the image contains critical information for answering the question) or irrelevant. This information is then fed to the VQA model to overcome language priors.

These above-discussed problems indicate that the empirical results of VQA systems do not reflect their efficacy, especially when promoting VQA systems to serve their intended purposes. Specifically, answering questions that the user cannot answer, such as in the healthcare domain. Therefore, it is important to make the output of the VQA model interpretable and not only rely on the evaluation results.

### 2.3. Interpretable VQA

In most real-world scenarios, human users want to get an explanation along with a VQA system’s output, especially if it fails to answer a question correctly or when the user does not know whether the answer is correct [6]. However, there exist only a few papers addressing the task of interpreting and explaining the outcome of VQA systems [2,5,6,7,40]. Also, most of the existing approaches rarely provide human-understandable explanations regarding the mechanism that led to a given answer. Li et al. [6] introduce a method that simulates the human question-answering behaviour. First, they apply pre-trained attribute detectors and image captioning to extract attributes and generate descriptions for the given image. Second, the generated explanations are used instead of the image data to infer an answer to a question. Consequently, providing critical attributes and captions to the end-user allows them to understand better what the system extracts from the image. Zhang et al. [5] introduce a heat map-based system to display the image’s regions relevant to the question to the user. To this end, they employed in their model region descriptions and object annotations provided in the Visual Genome dataset [41].

Pan et al. [7] introduce a method that provides counterfactual images along with a VQA model’s output. Precisely, for a given question-image pair, the system generates a counterfactual image that is minimally different from the original image and visually realistic but leads the VQA model to output a different answer for the given question. In its current form, their method is restricted to the context of colour questions. Furthermore, since their model makes edits on a pixel-by-pixel level, the counterfactual images contain changes also in areas irrelevant to a given question. Therefore, the counterfactual image does not provide any meaningful interpretability if the question-critical object has the same relative colour as other objects.

## 3. Method

In this paper, we propose a method named *COIN* that provides human-interpretable discriminatory explanations for VQA systems. The aim is to interpret the outcome of the VQA system by answering the question: “*How would the image look like so that the VQA system gives a different outcome?*”. Concretely, given an image-question pair (I;Q) and a VQA model f:(I,Q)→A, where *A* is its predicted answer, the goal is to train a model G to generate a new image $I^{\prime }$, such that:(1)$$G:\left(I,Q,A\right)\to I^{\prime }\;|\;f:\left(I^{\prime },Q\right)\to A^{\prime }\;;\;\forall A^{\prime }\neq A,$$
where $I^{\prime }$ is the counterfactual image of *I*. Since there are infinite possible images that can satisfy the above constraint, along the lines of Pan et al. [7], *COIN* aims to tackle the research question **RQ1** by constraining G to (i) be different as minimally as possible from *I*, (ii) be visually realistic, (iii) contain semantically meaningful edits and (iv) be applied only on the question-relevant regions. Intuitively, with these constraints, we aim to change the output of the VQA system by applying as minimum as possible changes only on the semantically relevant object so that the user can perceive what can change the output of the VQA system. To this end, we propose to extend the counterfactual GAN introduced by Pan et al. [7] by tackling, in addition to color-based questions, shape-based questions and ensuring that only the question-critical regions in an image are altered while retaining the rest of the image. The variables used for the system definition are summarized in Table 1.

Figure 1 illustrates an overview of the proposed architecture. In the depicted example, given the image *I*, the answer of the VQA system to the question “*Q: What color is the large central flower?*” is “*A: yellow*”. To explain this output, G goes through several components:

### 3.1. ROI Guide

To tackle the research question **RQ2**, G has to be guided to primarily edit regions in *I* that are relevant to *Q*. To this end, *COIN* aims to identify the question-critical ROI in *I*, but, complex images may contain various objects, of which, usually, only one or a few are relevant when answering a given question. Therefore, an object or a region can be considered to be question-critical if it is key to finding an answer to a given question. For example, given the question “*What colour are the man’s shorts?*”, the question-critical object in Figure 2 is the man’s shorts. Therefore, *COIN* aims to guide the generator with a continuous attention map $M\in R^{1\times h\times w}$ in the range [0, 1], where *h* and *w* correspond to the height and width of the input image, respectively. This map is supposed to highlight the discriminative ROIs of the image *I* that led f(I) to output the answer *A*. Thus, instead of generating a counterfactual image $I^{\prime }$ based on the original image *I*, the latter is concatenated with the attention map *M*, such that the concatenation [I;M] serves as an input to the generator G.

To obtain *M*, an attention mechanism is used to determine each pixel’s importance w.r.t the VQA system’s decision. The intuition is to identify the spatial regions in an image that are most relevant to answer a given question. For this reason, *COIN* applies the Gradient-weighted Class Activation Mapping (Grad-CAM) algorithm [43] to the VQA system’s final convolution layer because convolution layers retain spatial information that is not kept by fully connected layers. Specifically, CNN’s last convolution layer is supposed to have the finest balance between high-level semantics and fine-grained spatial information. Grad-CAM exploits this property by finding the gradient of the most dominant logit (i.e., in the case of a VQA system, this corresponds to the answer with the highest probability) that flows into the model’s final activation map. Furthermore, since Grad-CAM is suitable for various CNN-based models, it can be applied to most VQA systems.

Intuitively, the algorithm computes the importance of each neuron activated in the CNN’s final convolutional layer with respect to its prediction. Computing the gradient $y^{a}$ of the logit corresponding to the VQA system’s predicted answer *a* with respect to the *k*th feature map’s activations $\phi^{k}$ of a convolutional layer, i.e., $\frac{\delta y^{a}}{\delta \phi^{k}}$, reveals the localization map $L_{Grad-CAM}^{c}\in R^{u\times v}$ of width *u* and height *v*. Next, channel-wise pooling with respect to the width and height dimensions is applied to the gradients. The pooled gradients are then used to weigh the activation channels. Finally, the weighted activations $\alpha_{k}^{a}$ reveal each channel’s importance with respect to the VQA system’s prediction [43]:(2)$$\alpha_{k}^{a}=\frac{1}{Z}\sum_{i}^{u}\sum_{j}^{v}\frac{\delta y^{a}}{\delta \phi_{i,j}^{k}}$$

Performing a weighted combination of forward activation maps followed by a ReLU finally yields a coarse saliency map of the same size as the convolutional feature maps [43]:(3)$$L_{Grad-CAM}^{c}=ReLU\left(\sum_{k}\alpha_{k}^{a}\phi^{k}\right)$$

Finally, to obtain *M*, the feature maps $L_{Grad-CAM}^{c}$ are interpolated to match the size of the input image *I*. Furthermore, a gaussian filter with a mean μ=0 and a population standard deviation σ=2 is applied for improved preservation of the selected image regions’ edges [44].

### 3.2. Language-Conditioned Counterfactual Image Generation

To drive G to generate a counterfactual image $I^{\prime }$ such that the corresponding answer $A^{\prime }\neq A$, *COIN* follows Pan et al. [7] by adopting an architecture based on LingUNet [45], which is an encoder-decoder Neural Network (NN) similar to the popular pixel-to-pixel UNet model [46]. LingUNet maps conditioning language to key intermediate filter weights based on an embedding of natural language text.

Similarly to [7], *COIN* feed G with language embedding which is the concatenation of the VQA system’s question encoding and answer encoding. The question encoding is represented by the question embedding $\bar{q}$, which stems from the VQA system’s language encoding for the question *Q*. The answer encoding is represented by the answer embedding $\bar{a}$, which is the VQA’s final logits weight vector w.r.t its prediction *A* for the image-question pair (I,Q). The goal here is to train G with the VQA system’s negated cross-entropy for *A* being the target. Consequently, the generated image $I^{\prime }$ should contain semantically meaningful differences compared to *I*, such that the VQA system outputs two different answers for $I^{\prime }$ and *I* given the same question *Q*.

Precisely, G applies a series of operations to condition the image generation process on language. First, the question embedding $\bar{q}$ and the answer embedding $\bar{a}$ are concatenated to create a language representation $\bar{x}$. Second, G applies a 2D 1×1 convolutional filter with weights $K_{k}$ to each feature map $F_{j}$. Each $K_{k}$ is computed by splitting $\bar{x}$ into *m* equally sized vectors ${\left\{\bar{x}\right\}}_{j=1}^{m}$ and applying a 1×1 linear transformation to each of them. Applying the filter weights to each $F_{j}$ yields the language-conditioned feature maps $G_{j}$ [45].

Next, LingUNet performs a series of convolution and deconvolution operations to generate a new image $\hat{I}$. The final counterfactual $I^{\prime }$ is retrieved as follows:(4)$$I^{\prime }=M⊙\hat{I}+\left(1-M\right)⊙I,$$
where ⊙ denotes the element-wise multiplication and 1 is an all-ones matrix with the same dimension. Intuitively, $I^{\prime }$ is created by incorporating to the original image’s background, the foreground of $\hat{I}$, which is denoted by pixels with large attention values representing a higher intensity compared to those with low attention values.

### 3.3. Minimum Change

Although *I* and $I^{\prime }$ should have distinct semantics with respect to a given question Q, the differences between the two images should be as minimal as possible. To this end, *COIN* incorporates a reconstruction loss, which penalizes the generator for creating outputs different from the input. To ensure that question-critical objects can change their semantic meaning, the generator should be allowed to make significantly more changes in the corresponding image regions (i.e., the foreground) than in the rest of the image (i.e., the background). A modified $\ell_{2}$-loss adapted to this purpose, which incorporates the attention map *M* as a relative weighting term, acts as the reconstruction loss:(5)$$\ell_{2}=\left[\left|\right|\left(1-M\right)⊙I-\left(1-M\right){\left|\right|}_{2}^{2}\right].$$

Applying a weighted reconstruction loss aims at contributing to the desired traits that (i) the model predominantly edits critical objects and (ii) a relatively $\ell_{2}$-loss constraint is applied to question-critical regions, allowing for more significant semantic edits. Contrarily, the stricter $\ell_{2}$-constraint for question-irrelevant regions ensures that the generator retains them nearly unchanged.

### 3.4. Realism

The counterfactual images generated by G should be visually realistic. To this end and to tackle the research question (**RQ3**), *COIN* employs a PatchGAN discriminator as proposed by Isola et al. [47]. This discriminator learns to distinguish between real and fake images and penalizes unrealistic generated counterfactual images. The generator and the discriminator are trained in an adversarial manner as in GAN training [48].

#### Spectral Normalization for Stabilize Training

Training GANs can suffer from instability and be vulnerable to the problems of exploding and vanishing gradients [49]. In their approach, Pan et al. [7] applied gradient clipping to counter this problem, which requires extensive empirical fine-tuning of the training regime. To bypass this extensive procedure, *COIN* uses spectral normalization [49,50] to counteract training instability as Miyato et al. [50] suggest that using spectral normalization in GANs can lead to the generated images having a higher quality relative to other training stabilization techniques, such as gradient clipping. Given a real function γ:R→R, the Lipschitz constraint is followed if $|\gamma \left(x_{i}\right)-\gamma \left(x_{2}\right)\left|/|x1-x2\right|⩽k$, where *k* is the Lipschitz constanz (e.g., k=1). Given a CNN ${\mathrm{CNN}}_{\subseteq }$ with *L* layers and weights $\theta =\{w_{1},w_{2},\cdots ,w_{L}\}$, its output for an input *x* can be computed as [49]:(6)$${\mathrm{CNN}}_{\subseteq }=a_{L}⊙l_{w_{L}}⊙a_{L-1}⊙l_{w_{L-1}}⊙\cdots ⊙a_{1}⊙l_{w_{1}}\left(x\right),$$
where $a_{i=1}^{L}$ denotes the activation function in the *i*th layer. Spectral normalization regularizes the convolutional kernels $w_{i}\in R^{c_{out}\times c_{in}\times k_{w}\times k_{h}}$ with kernel width $k_{w}$ and height $k_{h}$ of the fully connected layers $l_{w_{i}}$ and $c_{in}$ and $c_{out}$ be the input and output channels, respectively. To this end, $w_{i}$ is first reshaped into a matrix ${\hat{w}}_{i}\in R^{c_{out}}\times \left(c_{in}\times k_{w}\times k_{h}\right)$, which is then normalized such that the spectral norm $\left|\right|{\hat{w}}_{i}{\left|\right|}_{sp}=1\forall i=1,\cdots ,L$. Thereby, the spectral norm is computed as follows [49]:(7)$$\left|\right|{\hat{w}}_{i}{\left|\right|}_{sp}=\frac{{\hat{w}}_{i}}{u_{i}^{T}\times {\hat{w}}_{i}\times v_{i}},$$
where $u_{i}$ and $v_{i}$ denote the left and sight singular vectors of ${\hat{w}}_{i}$ with respect to its largest singular value.

## 4. Experiments

In this section, we evaluate the effectiveness of the proposed approach from different aspects, namely, the capability of G to (i) generate counterfactual image $I^{\prime }$ such that $f\left(I^{\prime };Q\right)\neq f\left(I;Q\right)$ (**RQ1**), (ii) focus the changes on the question-critical region (**RQ2**), (iii) generate realistic images (**RQ3**). For result reproducibility and further improvements, we made our code and results publicly available under this link: coin.ai-research.net.

### 4.1. Dataset

For all our experiments, we used a subset of the VQAv1 dataset’s *Real Images* portion introduced by Agrawal et al. [3] The dataset covers images of everyday scenes with a wide variety of questions about the images and the corresponding ground truth answers. Despite the dataset includes samples with several types of questions, for feasibility reasons, we focus in this experiment on color- and shape-based questions only. This yields a set of 23,469 tuples (Image, Question, Answer). The subset is publicly available for further improvements, under: https://coin.ai-research.net/.

### 4.2. VQA System

In our experiments, we employed MUTAN [42], which is trained on VQAv1 dataset. MUTAN achieved overall accuracy of approximately 67% and it performed particularly well on questions with binary *Yes/No* answers (accuracy ≈85.14%). For quantitative questions (e.g., “*How many…?*”), it achieved and accuracy of 39.81%. For all other question types, including color and shape-based questions, it achieved an accuracy of 58.52%.

### 4.3. Evaluation and Results

Automatically and objectively assessing the quality of synthetically generated images is a challenging task [51,52,53]. Salimans et al. [52] suggest that there exists no objective function to assess a GAN’s performance. Furthermore, the goal of *COIN* is to provide an understandable interpretability to the VQA output. The quality of this interpretability can only be assessed by the satisfaction of the user. Therefore, we conducted a user study by presenting the output of *COIN* (i.e., $I^{\prime }$ and $A^{\prime })$ together with *I*, *A* and *Q* to the participants. For every sample, the user answers five questions divided in two phases:**Phase I:** We present the participant with $I^{\prime }$, *Q* and $A^{\prime }$. The participant is requested then to answer the following questions:*Is the answer correct?*with three possible answers: *Yes*, *No*, and *I am not sure*.*Does the picture look photoshopped: any noticeable edit or distortion (automatic or manual)?* with five possible answers (i.e., from Very real to Clearly photoshopped).**Phase II:** We present the participant with *Q* and both images *I* and $I^{\prime }$ together with the answers of the VQA system *A* and $A^{\prime }$. The participant is requested then to answer the following questions:*Which of the images is the original?* with three possible answers: *Image 1*, *Image 2* and *I am not sure*. Note that we do not indicate which of the images is *I* and which one is $I^{\prime }$*Is the difference between both images related to the question-critical object?* with three possible answers: *Yes*, *No* and *I am not sure**Which pair (Image, Answer) is correct?* with four possible answers: *Image 1* (Note that we do not which of the images is *I* and which one is $I^{\prime }$), *Image 2* (Note that we do not which of the images is *I* and which one is $I^{\prime }$), *Both* and *None*

To make this experience easy for the participants, we built a web application (coin.ai-research.net), where 94 participants have participated in the survey answering the above questions for 1320 unique samples. Note that some samples have been treated by more than one participant (maximum three), which make the total number of samples 2001. In the following, we present the qualitative and quantitative (obtained from the user study) results of *COIN* w.r.t the above-mentioned evaluation aspects:

#### 4.3.1. Semantic Change (RQ1)

The main goal of *COIN* is to interpret the result of VQA systems by trying to generate images with the minimum possible change from the original ones so that the VQA system changes its answer. Therefore, we evaluate here the capability of G to generate these images. Among 12,096 counterfactual images generated by G, 37.82% of them lead *f* to output an answer $A^{\prime }\neq A$. In particular, *f* outputs a different answer for 38.05% of the color-based questions and for 25.45% of the shape-based questions. This can be caused by several reasons, such as:The question-critical region is very large but the VQA system focuses on a very small region. Once altering that region, the VQA system slightly deviates its focus to another region (see the example in Figure 5b and result discussion in **RQ2**). Although the answer does not change, it interprets the outcome of the VQA system and its behaviour. Specifically, why the model outputs the answer *A* and whether the model sticks to a specific region for answering a question *Q*.The image requires a significant change so that the answer is changed but due to the other constraints (e.g., minimum change, realism, etc), the generator cannot alter the image more. Here also, the interpretation would be that the VQA system is confident about the answer and a lot of change is required to change its answer.The VQA system does not rely on the image while deriving the answer (see Section 2.2).

Among the samples treated in our survey, the participants found that the VQA outputs a correct answer *A* given *I* and *Q* for only 45.8% of the presented images, while it outputs a wrong answer for 41.4% of them. In the remaining 12.8% of the images, the participants couldn’t decide because (i) the question was not understood, (ii) the correct answer is not unique or (iii) the answer is only partially correct. After presenting the participants with both images *I* and $I^{\prime }$, the question *Q* and the answers *A* and $A^{\prime }$, the participants changed their opinions about 484 samples, where they found that *A* is correct for 70.5% of the presented images. This indicates that the participants could understand the question and answer better after interpreting the result of *f*. For the generated images, the participants found that *f* outputs a correct answer $A^{\prime }$ for only 40.5%.

Figure 3 illustrates qualitative results of G for color-based questions, where each row represents, from left to right, the original image *I*, the corresponding map *M* and the generated counterfactual image $I^{\prime }$ and its corresponding map $M^{\prime }$. The rows a–c of Figure 3 show examples of counterfactual images with different answers than their originals with realistic and understandable changes. As can be noticed in some examples such as Figure 3a,b, the VQA system *f* slightly shifts its attention after the change is applied. This means, that theoretically, *f* can give a different answer because of focusing on another region after the edit and not because of the edit itself.

For the rest of the samples, *f* fails to alter their semantic meaning. Figure 3d depicts such an example, where *f*’s answer does not change when being provided with the counterfactual image. The cause of this failure is due to the difference between the original and counterfactual images is too small for *f* to change the prediction. This because the clashing constraints that G has to obey. For example, G is required to change the output of *f* but is at the same time restricted to apply as minimally as possible of changes. Another reason might be a wrong gaudiness of the attention map. Suppose the question-critical object accounts for a large portion of the image, or the question is about the image’s background. In that case, G often only edits those parts on which the attention mechanism focuses. As a result, the relevant image region is not modified in its entirety and thus *f* cannot perceive a semantically meaningful change or its attention is shifted towards other regions of the question-critical object.

While G can generate semantically meaningful counterfactual images for a lot of color-based questions, it is not the case for shape-based questions as shown in Figure 4. While the VQA system predicts a different answer in both cases, the changes are not semantically meaningful from a human observer’s perspective. In Figure 4a, the object’s shape remains roughly unchanged, while the counterfactual generator slightly edits the sign’s color. However, the task of G is accomplished, where the interpretation is that the VQA system is prone to any small change in the input to generate a different answer. Furthermore, *f* predicts an incorrect answer for both the original and the counterfactual image. As *M* indicates, the changes from the original image seem to be significant enough for *f* to shift its focus slightly to the lower right portion of the sign when making an inference on the counterfactual image.

This shift seems to drive the model to change its prediction. Contrarily, the changes in Figure 4b are more dominant, where G produces an artifact covering the kite and a segment of the sky surrounding it. As *M* shows, *f* focuses on the same area in both the original and the counterfactual image, but the artifact seems to be the cause of the different answer. These two instances are exemplary for most of the counterfactual images for shape-based questions: the generator (i) applies only a few edits that are barely noticeable but they are more likely to change the answer of *f* which is the task of G or (ii) produces artifacts that are not semantically meaningful for a human observer.

Both the examples in Figure 3 and Figure 4 show that G’s edits vary depending on the questions and answers. Since the attention maps pose a strong constraint for G, its edits heavily depend on them. If *M* focuses on the question-critical object, such as in Figure 3c, G successfully modifies it. Contrarily, if *M* focuses only on a small portion of the object/region of interest (such as in Figure 3e), the language conditioning does not have the desired effect. In these cases, G fails to modify the areas relevant to the question-answer pair sufficiently. Despite the failure to generate the counterfactual image, G provides an understandable interpretability to the behaviour and result of the VQA system for a particular pair (Image, Question).

#### 4.3.2. Question-Critical Object (RQ2)

One of the main aims of *COIN* is to edit only the question-critical object. This is controlled by the attention map *M*, where the generator G is restricted to apply changes predominantly in the regions of the image on which *M* focuses. Figure 5 depicts, for three example samples, the original image *I* (Left), the question *Q*, the answer *A* given by MUTAN (*f*) and the interpolation of the map *M* with *I* (Center). The right image is the *background* obtained by computing (1−M)⊙I, where 1 denotes all-ones matrix and ⊙ is the element-wise multiplication.

In Figure 5a, the question is about the ball and as shown in the interpolated map, *f* focuses specifically on the ball’s region, which makes G restricted to make changes only on that region. When the region of interest is large and/or sparse such as in Figure 5b, *f* might not focus on the entire question-critical region but only a portion of it, which is sufficient to answer the question. Figure 5c indicates that *f* wrongly answered the question as *red and white* but the correct answer is clearly *black*. The obtained map *M* explains that *f* was focusing on a different region. Consequently, G applies the changes on the wrong region. The user can understand the behaviour of the model based on this generated image, which is an edit to the original one w.r.t a wrong region.

According to our user study, G applied the changes on the critical object in 50.1% of the samples. For 32.2% of the samples, the changes were applied on (1) completely different region, (2) the region of interest and other irrelevant regions or (3) a small part of the region of interest. For the remaining 17.7%, the users couldn’t determine whether the changes were applied on the question-critical object or not. From these results, we can derive three main patterns of the attention mechanism:If the object relevant to answering a question is relatively small compared to the rest of the image, the attention mechanism focuses on it completely in most cases. In other words, the computed intensities are higher for pixels belonging to the object than for the rest of the pixels. Under these circumstances, the generator can make larger changes to the entire object than to the rest of the image.Contrarily, if the object is very large or MUTAN pays attention to the background, the projection usually focuses only on a part of it. Consequently, the information that the generator receives allows it to apply more significant changes to a segment of the object or the background than to the rest of it.If MUTAN makes an incorrect prediction, this is often reflected by the projection not focusing on the question-critical object, but another element of the image, such as in Figure 5c.

In all these patterns, the applied changes of *I* w.r.t *M* is supposed to change the answer of *f* for the same question *Q*. This is because *M* is indicating where *f* is focusing to answer *Q* given *I*. However, when the VQA focuses on an irrelevant region in *I*, the applied changes might change the visual semantic of this irrelevant region such that when feeding *f* with $I^{\prime }$ and *Q*, *f* focuses on a different region than it did in *I*. This different region can also be the correct region.

#### 4.3.3. Realism (RQ3)

Generating realistic counterfactual images is very important to interpret the result of VQA systems to users. As shown in Figure 3 and Figure 4, the degree of realism varies depending on the necessary edit that changes *f*’ answer and on the size difference of the question-critical region to the focused object. This is reflected also in the result of our user study that is demonstrated in Figure 6, where the users were presented only with the generated images and asked “*Does the picture look photoshopped (any noticeable edit or distortion)?*”. As Figure 6 indicates, the answers of the users vary depending on the generated images. When presenting the corresponding original image together with the generated one and asking “*Which of the images is the original?*”, the participants could correctly distinguish between the original and the generated one in ∼67.2% of the presented samples. In ∼12.9% of the images, the participants selected the generated image as the original one and in the rest of image (∼19.9%), the participants couldn’t decide which of the images is the original and which one is the generated. This result indicates that G could trick the human participants by generating counterfactual which look extremely realistic.

#### 4.3.4. Minimality of Image Edits

To evaluate G’s performance on generating counterfactual images with minimum edits, we computed $\ell_{1}$-norm across both the training and the validation set. This measures the magnitude of changes in the counterfactual image relative to the original one, where lower values indicate fewer changes.

Table 2 summarizes the results of this evaluation, where the mean (denoted μ) and standard deviation (denoted σ) values are calculated for different splits of both the training and the validation sets. The first three columns represent the values computed across the entire dataset and for color-based and shape-based questions. The remaining six columns contain the same computations for the portion of pairs of original and counterfactual images for which G predicts distinct or equal answers, respectively.

The results of Table 2 indicate that G applied fewer changes when it comes to color-based questions compared to shape-related questions. This observation applies to both the training and the validation set and across all splits. Moreover, overall, G applied fewer changes in those cases where *f*’s predictions regarding the original and counterfactual image were distinct than if they were equal. This indicates that G changed the original image to the maximum possible level but without successfully changing *f*’s answer.

## 5. Conclusions

In this paper, we introduced *COIN*, a GAN-based approach to interpret the output of VQA models by generating counterfactual images to drive the VQA model outputting different a different answer (**RQ1**). *COIN* is a modified implementation of LingUNet with incorporating a Grad-CAM-based attention mechanism that determines each pixel’s importance regarding the VQA model’s decision making process. With this, the counterfactual generator learns to apply modifications in an image predominantly to question-critical objects, while retaining the rest of the image (**RQ2**). The obtained results indicate that using an attention mechanism is an appropriate means to guide the modification process. Furthermore, the quality of the counterfactual images depended to a large extent on the attention maps. Extensive experiments on the challenging VQAv1 dataset have demonstrated that *COIN* achieves encouraging results for color questions by generating realistic counterfactual images (**RQ3**).

For future work, we will train *COIN* on a larger, more diverse dataset such as VQAv2 dataset, which contains multiple images per question rather than only a single one as in the VQAv1 dataset. Moreover, using an attention mechanism that focuses on the question-critical objects more accurately could also significantly improve the interpretability capabilities of *COIN*. To this end, we plan to employ super-pixel segmentation to extract concepts (e.g., color, texture, or a group of similar segments) and uses the Shapley Value algorithm to determine each concept’s contribution to a DNN’s decision. The generator will then be trained to alter the most important concept(s) in an image rather than providing it an attention map. Moreover, replacing an entire instance of a concept rather than editing an image on a pixel-by-pixel level could pave the way for semantic changes even larger than altering shapes.

## Figures and Tables

**Figure 1 sensors-22-02245-f001:**
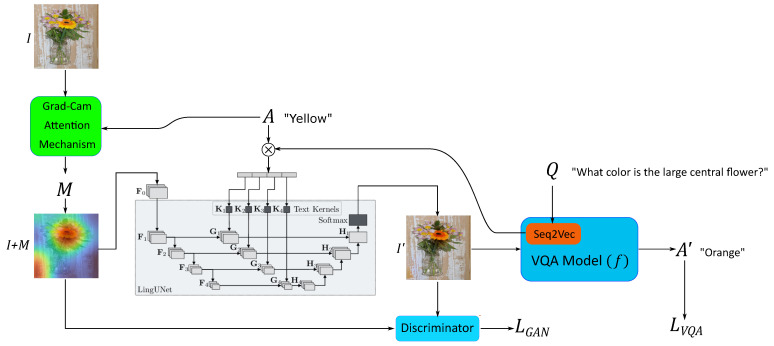
Overview of the proposed architecture inspired by Pan et al. [7].

**Figure 2 sensors-22-02245-f002:**
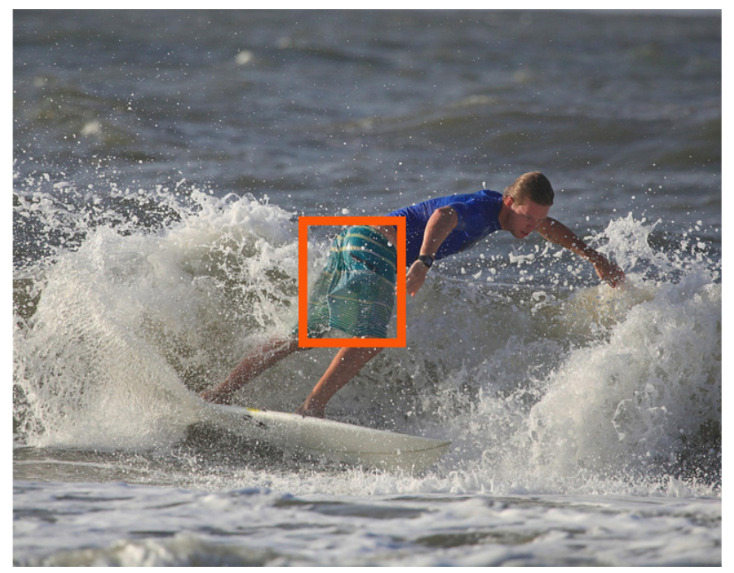
Example question-image pair from the VQAv1 dataset [3]. The red bounding box indicates the question-critical object.

**Figure 3 sensors-22-02245-f003:**
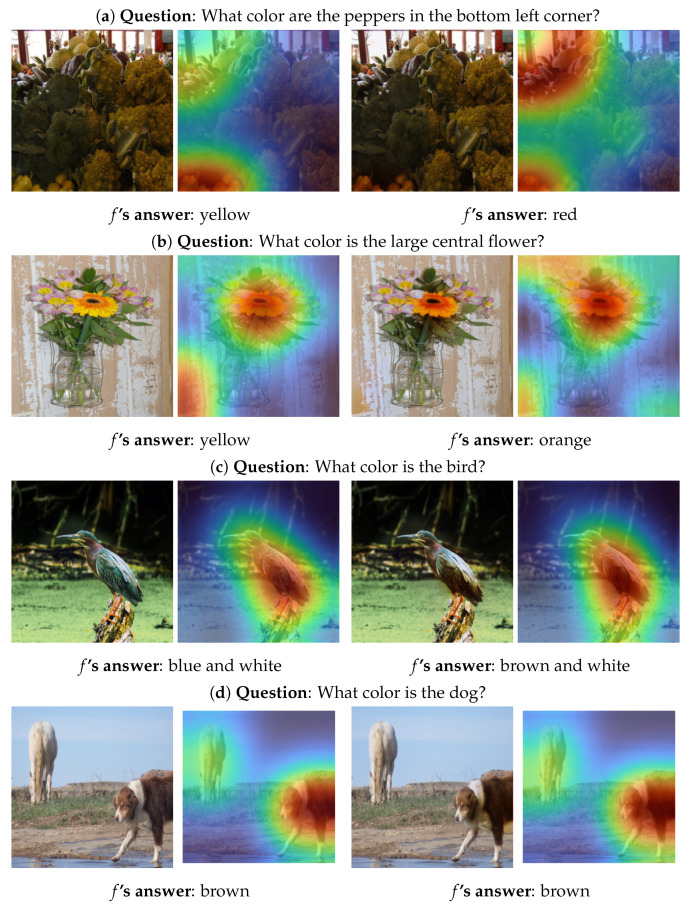
Examples of G’s output for color-based questions from the VQAv1 [3] validation set. Left: original image *I* and the corresponding attention map *M*. Right: Generated counterfactual image $I^{\prime }$ and the corresponding attention map $M^{\prime }$.

**Figure 4 sensors-22-02245-f004:**
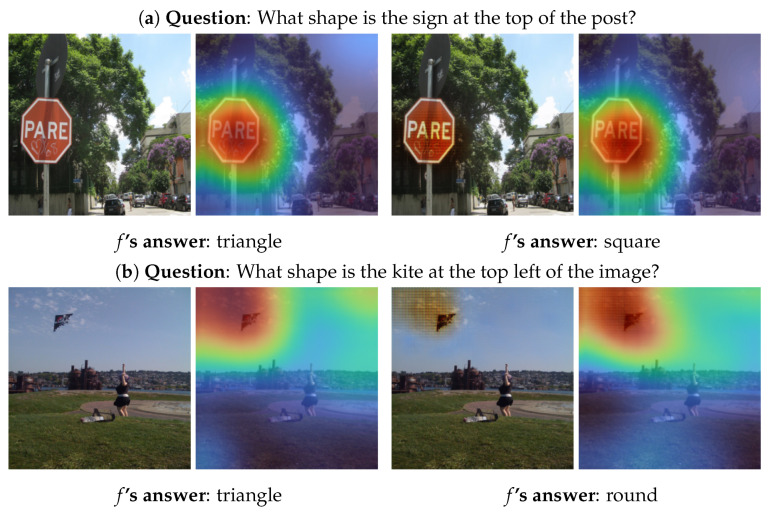
Examples of G’s output for shape-based questions from the VQAv1 [3] validation set. Left: original image *I* and the corresponding attention map *M*. Right: Generated counterfactual image $I^{\prime }$ and the corresponding attention map $M^{\prime }$.

**Figure 5 sensors-22-02245-f005:**
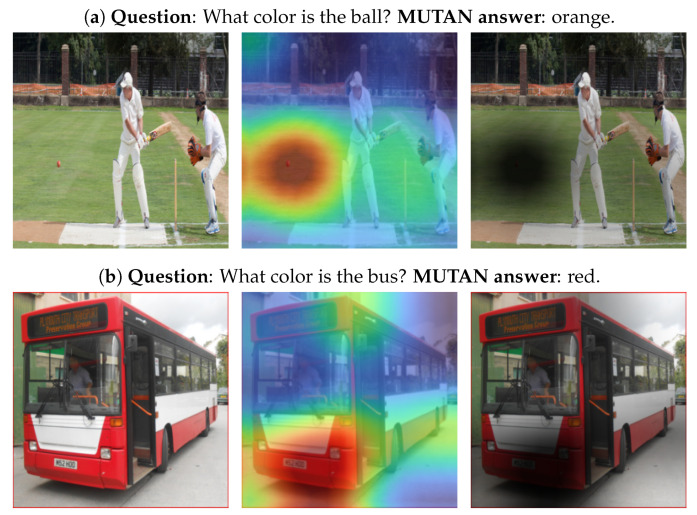
Example outputs of the Grad-CAM algorithm applied to MUTAN for color-based questions. Left: original image. Center: Interpolated attention map projected on the original image. Right: The background image.

**Figure 6 sensors-22-02245-f006:**
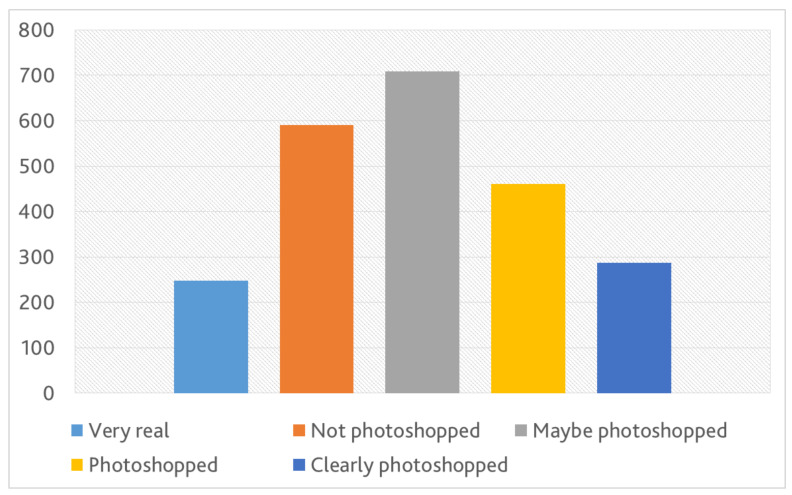
Frequency histogram of participants’ answers to the question: “*Does the picture look photoshopped?*”.

**Table 1 sensors-22-02245-t001:** Summary of variables used in this paper.

Variable	Description
G	The counterfactual generator proposed in this paper.
*f*	The VQA system (i.e., MUTAN [42] in this paper)
*I*	Original image
*Q*	Question about *I*
*A*	The answer of *f* to *Q* given *I*
*h*	*I*’s height
*w*	*I*’s width
$I^{\prime }$	The counterfactual image of *I*, generated by G
$A^{\prime }$	The answer of *f* to *Q* given $I^{\prime }$
$\hat{I}$	An image generated by G
*M*	The attention map of *I*
$M^{\prime }$	The attention map of $I^{\prime }$

**Table 2 sensors-22-02245-t002:** Mean (μ) and standard deviation (σ) of the $\ell_{1}$-norm computed across the training and validation set and split across categories.

		Training Set		Validation Set	
		μ	σ	μ	σ
All		0.0175	0.0039	0.0175	0.0041
Color		0.0174	0.0039	0.0174	0.004
Shape		0.0177	0.0048	0.0208	0.0047
Same VQA Answers	ALL	0.0207	0.0040	0.0177	0.0041
	Color	0.0176	0.0039	0.0176	0.0041
	Shape	0.0212	0.0049	0.0212	0.0048
Different VQA Answers	ALL	0.0173	0.0039	0.0173	0.0038
	Color	0.0172	0.0038	0.0173	0.0038
	Shape	0.0195	0.0046	0.0198	0.0042

Bold is used to highlight the best/worst result.

## Data Availability

All details of the conducted experiments and used data can be found under the link: coin.ai-research.net.

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
