# Peer review of "COIN: Counterfactual Image Generation for Visual Question Answering Interpretation"

_sensors, 2022, doi:10.3390/s22062245_

Round 1

Reviewer 1 Report

This paper provided a COIN mechanism that generated counterfactural images for more interpretation in VQA models. There are three aspects to analysis the results of COIN including semantic(RQ1), Question-critical object(RQ2), Realism(RQ3). However, the experiment results seem not good. The results do not compare with state-of-art and more explanation need to be presented for the contribution.

  1. The goal of Coin is to generate a conterfactural image with different answer in VQA model. In Line394, only 37.82% of counterfactural images have different answer in RQ1. However, the goal of COIN is to generated the image with different answer.

  1. In the figure 4, Only show the bad shape-based question results. The results show the COIN can’t deal with shape-based question in RQ1.

  1. According to the study, the changes that is on the critical object is only 50.1%. However, the aims of COIN are to edit only the question-critical object. With the technology of object detection in state-of-art, there are large improvement space.

  1. In figure 6, Only less than half generated images are be considered as not photoshopped or very real. 2% participants can distinguish original and generated one. The results seem the generated image is not real.

There are some typos needs to correct. The first occurrence of a proper noun should display the full text.

The contribution of the paper needs to be clear stated and compared with state-of-art.

Author Response

Dear reviewer, 

First, we would like to thank you for carefully reading our paper and providing us with valuable feedback. We revised the paper according to your comments and in the following, we answer each of them: 

Reviewer1: “The goal of Coin is to generate a conterfactural image with different answer in VQA model. In Line 394, only 37.82% of counterfactural images have different answer in RQ1. However, the goal of COIN is to generated the image with different answer.

Authors' answer: We agree with the reviewer that the generator did not change the answer of the VQA model of more than 60% of the images. However, the main goal of the COIN is not to change the answer of the VQA model but to interpret its outcome. Changing the answer is rather the main way to accomplish this task. As mentioned in the sections: “Experiments→Semantic change (RQ1)” and “Experiments→Question-critical object (RQ2)”, the generator cannot change
the answer for several reasons, such as:

  • The question-critical region is very large but the VQA model focuses on a very small region. Once changing that region, the VQA model slightly deviates its focus to another region (see the example in Figure 5.b). Although the answer does not change, it interprets the outcome
    of the VQA model and its behaviour. Specifically, why the model outputs the answer A and whether the model sticks to a specific region for answering a question Q.
  • The image requires a significant change so that the answer is changed but due to the other constraints (e.g. minimum change, realism, etc), the generator cannot alter the image more. Here also, the interpretation would be that the VQA model is confident about the answer and
    a lot of change is required to change its answer.
  • The VQA model does not rely on the image while deriving the answer (discussed in the introduction and related works as an inherent problem of VQA systems in general). As mentioned in Section Semantic change, the participants found that the VQA model generates a correct answer only for 40.5% of the generated images. This indicates that any change in the image couldn’t change the answer of the VQA model.

Reviewer1: “In the figure 4, Only show the bad shape-based question results. The results show the COIN can’t deal with shape-based question in RQ1

Authors' answer: Again the goal of COIN is to interpret the outcome of the VQA model. Figure 4 shows that any small change (sometimes not even noticeable by the naked eye) in the image can change the answer of the VQA model (what the generator is supposed to do in the first place). In both examples, the VQA answer has been changed demonstrating that VQA models are still struggling
with some questions such as shape-based questions.

Reviewer1: “According to the study, the changes that is on the critical object is only 50.1%. However, the aims of COIN are to edit only the question-critical object. With the technology of object detection in state-of-art, there are large improvement space.

Authors' answer: Here, we would like also to emphasize on the goal of the paper is interpreting the outcome of VQA models and not just changing the answer. For every image, we would like to show to the user why the VQA model provides an answer A. Therefore, we use the attention map which is derived from the last convolutional layer of the VQA model. Applying object detection would
defiantly be much simpler but it does not serve the purpose of the paper. Note that we do not assume at any time that the VQA model is giving a correct answer neither for the original image nor for the generated one. We are planning on improving the method by combining super-pixel segmentation with an attention map to alter the entire region (although the VQA model derives
its answer from only a smell region) so that even if the VQA model deviates its focus, it will find the region already changed. However, we don’t assume that this improves the interpretability but only the ”semantic change”.

Reviewer1: “In figure 6, Only less than half generated images are be considered as not photoshopped or very real. 2% participants can distinguish original and generated one. The results seem the generated
image is not real.

Authors' answer: Figure 6 shows the answers of all participants for all images. It is completely normal that the participants can distinguish between original and generated images as the goal of generating images is not to make them look more realistic than the original ones1 but to make them look realistic to a certain extent so that the user can understand the outcome of the VQA model.

Reviewer1: “There are some typos needs to correct. The first occurrence of a proper noun should display the full text.

Authors' answer: Thank you for drawing our attention to them. We re-revised the paper and corrected the typos.

Reviewer1: “The contribution of the paper needs to be clear stated and compared with state-of-art.

Authors' answer: We re-formulated the contribution of the paper. Regarding comparing it with state-of-the-art, we believe that interpretability is still a challenging task in Machine Learning and up to our knowledge, it is still impossible to compare its results as they are assessed only by the satisfaction
of the users. However, we would be happy to conduct any comparison if the reviewer believes that we are not aware of a comparison scheme. 

Sincerely yours,

Zeyd Boukhers, Timo Hartmann and Jan Jürjens

Reviewer 2 Report

First of all, I should say that visual question answering is a hot research topic within artificial intelligence now. I absolutely believe this manuscript will be interesting to the readers and can attract much attention. The authors conducted modern experiments and gave answers to several important research questions related to the future perspectives on the road to general artificial intelligence. I have no comments on the methodology and experiments conducted, but despite my confidence that the manuscript can be accepted without major revision, I ask the authors to rethink the many abbreviations in the title and keywords. I suppose the avoiding of abbreviation VQA in the title which can not be qualified as generally accepted and changing it to the full Visual Question Answering should reflect the main text more correctly. Once again this note needs no major revision of the whole manuscript and if the authors agree with my opinion then they can accept my comment by changing the title.

Author Response

Dear reviewer, 

First, we would like to thank you for carefully reading our paper and providing us with valuable feedback. We changed the title according to your comment and mentioned all the acronyms with their full meaning in their first appearances.

Sincerely yours,

Zeyd Boukhers, Timo Hartmann and Jan Jürjens

Round 2

Reviewer 1 Report

I think my major concerns have been correctly addressed by detailed illustration.  I would like to thank the authors for their revision, which has improved the manuscript.